# Spatial navigation with horizontally spatialized sounds in early and late blind individuals

**Samuel Paré** [ID] [1◉], **Maxime Bleau** [ID] [1◉], **Ismaël Djerourou** [1◉], **Vincent Malotaux** [2‡], **Ron Kupers** [1,2,3‡], **Maurice Ptito** [1,3‡] *

**1** École d'Optométrie, Université de Montréal, Québec, Canada, **2** Institute of Neuroscience, Université Catholique de Louvain, Brussels, Belgium, **3** Institute of Neuroscience and Pharmacology (INF), University of Copenhagen, Copenhagen, Denmark

◉ These authors contributed equally to this work.
‡ These authors also contributed equally to this work
* maurice.ptito@umontreal.ca

**Data Availability Statement:** All relevant data are within the manuscript and its Supporting Information files.

**Funding:** The study was funded by the Harland Sanders Chair in Visual Science and the Synoptik

## Abstract

Blind individuals often report difficulties to navigate and to detect objects placed outside their peri-personal space. Although classical sensory substitution devices could be helpful in this respect, these devices often give a complex signal which requires intensive training to analyze. New devices that provide a less complex output signal are therefore needed. Here, we evaluate a smartphone-based sensory substitution device that offers navigation guidance based on strictly spatial cues in the form of horizontally spatialized sounds. The system uses multiple sensors to either detect obstacles at a distance directly in front of the user or to create a 3D map of the environment (detection and avoidance mode, respectively), and informs the user with auditory feedback. We tested 12 early blind, 11 late blind and 24 blind-folded-sighted participants for their ability to detect obstacles and to navigate in an obstacle course. The three groups did not differ in the number of objects detected and avoided. However, early blind and late blind participants were faster than their sighted counterparts to navigate through the obstacle course. These results are consistent with previous research on sensory substitution showing that vision can be replaced by other senses to improve performance in a wide variety of tasks in blind individuals. This study offers new evidence that sensory substitution devices based on horizontally spatialized sounds can be used as a navigation tool with a minimal amount of training.

## Introduction

Vision is the most important aspect for spatial navigation and mobility in humans; it is constantly used for movement guidance, route planning and orientation [1–3]. Therefore, visually impaired individuals face several challenges when navigating, such as disorientation, detecting and avoiding obstacles [4]. Thanks to the long cane and guide dogs, some of these issues can be addressed. For instance, ground based obstacles can be detected by the cane or avoided by the guide dog but obstacles above waist remain problematic [5]. Nevertheless, blind individuals lack navigational independency [6]. For this reason, sensory substitution devices (SSDs)

Foundation in Denmark. The funders had no role in study design, data collection and analysis, decision to publish, or preparation of the manuscript.

**Competing interests:** The authors have declared that no competing interests exist.

[7,8] have been developed to bring visual information through other sensory modalities [9,10] such as audition [11] or touch [12,13].

The vOICe is one of the best studied visual-to-auditory (VTA) SSD. The camera of the device scans its field of view from left to right, thereby offering momentary "snapshots" of the environment in the form of sound cues. The vOICe informs the user about the vertical and horizontal positioning of objects, as well as brightness of the environment. Vertical and horizontal positioning are indicated by the frequency and the length of the sound, respectively, whereas brightness is indicated by differences in amplitudes of the sound oscillations. The vOICe demands the user to analyze multiple spectral cues, with a two second delay between each scan, to extract important information, detect and identify objects [11].

The Tongue Display Unit (or TDU) is a tactile-to-vision (TTV) SSD capable of transmitting images to the tongue in the form of electrotactile pulses. The TDU is composed of a tongue array consisting of 400 small circular electrodes arranged in a 20x20 matrix, a computer and a webcam. Every time an object enters within the visual field of the camera, the visual image is translated into electrotactile pulses that are transmitted to the tongue through the electrode array. The obstacles are thus 'drawn' with electrical current on the tongue in real time from the images provided by the camera [14–16].

The Sound of Vision (SoV) is a more recently developed VTA SSD. The SoV provides combined audio and tactile feedback by using multiple cameras and depth sensors that are worn on the forehead and which are connected to a laptop stowed in a backpack and worn by the user. The system informs the user of obstacles positioning with vibrations to the abdomen through a haptic belt. Then, the SoV conveys depth information (or overall 3D objects' shape) to the user by translating all 3D points into binaural sound effects of "popping bubbles" that will be modulated in loudness and pitch for proximity and elevation respectively [17].

These devices aim to give the user a "visual-like" experience of the environment by giving information about the whole visual field of the camera. In these cases, the user must analyze all the information given by the SSDs to extract what is useful for the task to be accomplished. While studies have shown that it is possible to achieve proficient levels of navigation with these devices in laboratory environments, such performance often requires an intensive training period [15,18–21]. Indeed, learning to use these devices is a process that imposes a heavy load on cognitive resources, often creating a feeling of exhaustion [7,22].

There is hence a need for simpler SSDs that provide more pertinent information needed for a specific task. To be efficient as navigation aids, SSDs should provide information relevant for navigation by processing the raw visual data, extracting only spatial information (i.e. depth and position) and delivering it in a simple and meaningful format. This would minimize the strain on attentional resources, and therefore shorten the user's reaction time [22]. In order to deal with some of the SSDs' shortcomings, Maidenbaum and colleagues developed the Eye-Cane device, a more "user-friendly" mobility aid that provides "point-to-distance" information. In short, the EyeCane detects tangible obstacles with an infrared light sensor and is able to calculate the distance between the device and the detected object, providing this sole information in the form of vibration: the higher the vibration, the closer the object [23]. As a result, the user can navigate through obstacles by scanning the environment with the device [21,24,25].

In this pilot study, we tested the *Guidance-SSD* (GSSD), a newly developed smartphone application that expands on the concept of "point-to-distance" introduced with the EyeCane [23]. This application can be installed on a regular smartphone, thereby taking advantage of its cameras and information processing capacity. Instead of vibrations, the GSSD conveys information about all tangible obstacles in the environment by using horizontally spatialized sounds, with the aim to provide "guidance" to the user. Here, we define horizontally spatialized

sounds as the combined auditory cues that allow the localization of objects (regardless of their height) on the horizontal plane. To do so, these auditory cues represent two relevant spatial inputs: 1) the object's distance from the user and 2) its direction. Therefore, we hypothesize that when confronted with obstacles in their path of travel, participants using the application should have enough information to be "guided" through obstacles, hence the GSSD name.

To test this hypothesis, we investigated if participants could detect and avoid obstacles in a life-size obstacle course using this new application to guide their movements. We included early blind (EB), late blind (LB) and blindfolded sighted control subjects (SC) in a detection and avoidance task. We hypothesized that participants would be able to perform both tasks above chance level with a minimum amount of training. Moreover, given the established spatial auditory capacities of the blind [13,26], we supposed that both blind groups would outperform SC and that EB would be better than LB.

## Methodology

### Participants and ethics

A total of 12 EB (mean age: 45 ± 10; 5 women), 11 LB (mean age: 40 ± 12, 8 women) and 24 SC (mean age: 40 ± 12, 11 women) participated in the study. Not all participants completed both parts of the study (3 EB, 1 LB and 7 SC did not complete the navigation task while 3 LB and 1 SC did not complete the detection task for reasons of time and availability). Participants were recruited from the Institut Nazareth et Louis-Braille (INLB) in Montreal (Canada) and the BRAINlab of the University of Copenhagen (Denmark). Age and sex-matched control subjects were also recruited from the Montreal and Copenhagen area. All LB participants had acquired blindness at the age of 16 or older. To evaluate the influence of experience-dependent plasticity in the LB group, we calculated the blindness duration index (BDI) according to the formula "(age-age onset blindness)/age" (as described in [21]). The BDI score can vary from 0 to 1, expressing the relative amount of time a person has been blind, with low scores indicating recent onset of blindness and high scores long duration of blindness. The average BDI was 0.52 ± 0.16 (range: 0.13 till 0.66) while the mean onset of blindness was 21.4 ± 6.6 years. All blind participants were users of the long cane, while three of them mainly used a guide dog. None of the participants had associated neuropathy, hearing loss or other pathology that could affect navigation performance and mental spatial representation. Demographic data of the blind participants can be found in Table 1. All participants were blindfolded during the experiment. The experimental protocol was approved by the Comité d'éthique de la recherche Clinique de l'Université de Montréal (CERC-19-097-P) and all participants provided written informed consent before the experiment.

### Apparatus

The navigation system uses a Lenovo Phab 2 Pro smartphone, bone conducting headphones and a headset supporting the smartphone at eye-level (see Fig 1A). The smartphone is equipped with a Qualcomm Snapdragon 652 processor, a 4050 mAh battery, 4 GB of RAM, and uses Android 6.0. The smartphone has an RGB camera, a depth camera and a motion tracking fisheye camera. For this experiment, the phone is placed in a custom head mount adjusted so the cameras are at eye-level (see Fig 1D).

The GSSD uses horizontally spatialized sounds to convey visual information to the user. Using the phone's cameras, the system can detect tangible objects within a 3-meter radius in front of the user and signal their location in the horizontal plane through sonification. The auditory signal encodes the objects' polar coordinates with the user's position as the point of origin. To convey the angular coordinate, or the objects' azymuth in relation to the user, the

**Table 1. Blind participants' characteristics.**

| Participants | Age & Sex | Blindness Onset (years) | BDI[a] | Cause | Residual Perception | Participants | Age & Sex | Blindness Onset | Cause | Residual Perception |
|---|---|---|---|---|---|---|---|---|---|---|
| **LB1**[*] | 55W | 24 | 0.56 | RP[b] | LP[j] | **EB1**[**] | 56M | Perinatal | ROP[i] | - |
| **LB2** | 25M | 17 | 0.32 | RP | - | **EB2** | 49W | Perinatal | ROP | - |
| **LB3** | 50M | 17 | 0.66 | A[e] | - | **EB3** | 44W | Perinatal | ROP | - |
| **LB4** | 44W | 17 | 0.61 | GL[f] | - | **EB4** | 31W | Perinatal | ROP | - |
| **LB5** | 56W | 20 | 0.64 | RC[g] | - | **EB5** | 18M | Perinatal | ROP | - |
| **LB6** | 47W | 22 | 0.53 | DR[h] | - | **EB6** | 49M | Perinatal | ROP | - |
| **LB7** | 44W | 17 | 0.61 | GL | - | **EB7** | 33M | Perinatal | ROP | - |
| **LB8** | 56W | 20 | 0.64 | RC | - | **EB8** | 49W | Perinatal | ROP | - |
| **LB9** | 47W | 22 | 0.53 | DR | - | **EB9** | 34W | Perinatal | ROP | - |
| **LB10** | 38W | 20 | 0.47 | GL | - | **EB10** | 46M | Perinatal | ROP | - |
| **LB11** | 46M | 40 | 0.13 | M[d] | - | **EB11** | 49M | Perinatal | ROP | - |
| | | | | | | **EB12** | 28M | Birth | LA[c] | LP |

[*]**LB**: Late blind

[**]**EB**: Early blind

[a]**BDI**: Blind duration index

[b]**RP**: Retinitis pigmentosa

[c]**LA**: Leber's amaurosis

[d]**M**: Meningitis

[e]**A**: Accident

[f]**GL**: Glaucoma

[g]**RC**: Retinal cancer

[h]**DR**: Diabetic retinopathy

[i]**ROP**: Retinopathy of prematurity

[j]**LP**: Light perception.

system uses binaural differences, thus creating a possible 360-degree audio feedback. To signal the radial coordinate, or the distance between the object and the user, the system uses a combination of three previously tested sonification strategies [27] [: 1. Beep Repetition Rate (BRR), where the interval between each beep correlates positively with distance; 2. Sound Fundamental Frequency (SFF), whereby the sound frequency (pitch for the user) correlates negatively

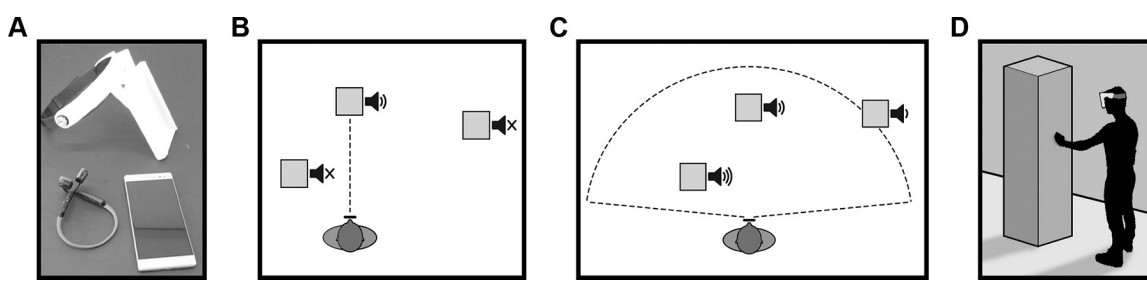

**Fig 1. The apparatus and the two modes offered by the system.** (A) The custom head mount, the bone-conducting headphones (lower left) and the Lenovo Phab 2 Pro smartphone. (B) In detection mode, auditory feedback is given about one point in space. The system indicates the presence of a tangible object in a straight line in front of the user (angle: 0-degree, range: 3 meters). (C) The avoidance mode's feedback consists in a 3D audio construct of the environment; it detects everything in the fisheye camera's field of view and renders every tangible surface location in relation to the user (range: 3 meters). Squares represent obstacles, speaker icons illustrate the sound heard by the participant. The more bars there are, the higher the BRR, SF and SI. An X illustrates the absence of sound. (D) A blindfolded participant using the device to reach an obstacle. The device is head-mounted so the camera is at eye-level and facing towards the environment in front of the participant.

with distance; and 3. Sound Intensity (SI) which also correlates negatively with distance. Thus, when the user approaches an obstacle, the combination of the three components (the time between the beeps shortens while the pitch and the SI get higher) allows the user to estimate the distance accordingly.

The sonification used in this study relies on synthetic liquid sound effects following the proposition of Doel [28]. It is said that liquid sounds are prevalent in our environment and are easily identified, making them suitable to represent spatial information when simulated [28]. Based on that, the GSSD simulates water droplets sounds as the individual beeps in the signal and synthetically modulates these sounds' properties according to the three sonification strategies (BRR, SFF and SI).

The GSSD software offers a *detection* and *avoidance* mode. Both modes use horizontally spatialized sounds but in two different ways. The detection mode offers only one sound source, which is a straight line directly in front of the user (0-degree angle). Thus, this mode provides "point-to-distance" information similarly to the EyeCane device [23] and should be used in a comparable fashion: the user obtains information about obstacles by scanning (or pointing) the environment. However, the avoidance mode uses the cameras' whole field-of-view (170 degrees) to convey information on location for every tangible object that could be in the user's path of travel. Each object is represented by one sound source representing its closest point or edge in relation to the user. Thus, the user can simultaneously hear multiple objects and plan his/her movements to avoid collisions. With the Avoidance mode, participants could detect a maximum of 3 obstacles simultaneously (both walls and an obstacle) because of the design of the corridor. Obstacles behind the user are no longer sonified to avoid cognitive overload.

## Experimental walkway

The life size obstacle course consisted of a corridor (21m long, 2.4m wide) where six obstacles were placed 3 meters apart from each other on the longitudinal axis, but randomly placed on the horizontal axis. The obstacles were made of cardboard boxes (L: 0.45m; l 0.4m; H: 1.9m) to avoid injury on impact.

## Tasks

Navigation refers to the capacity to move in space, to orient oneself in space and avoid unexpected obstacles. Consequently, navigation encompasses multiple cognitive processes [29,30]. Furthermore, obstacle avoidance requires not only to detect obstacles, but also to judge and memorize the obstacle's location, to estimate distance, to adjust the path of travel accordingly and to get back to the initial planned route in order to reach a destination safely. In other words, a navigation task necessitates higher cognitive spatial processing than an obstacle detection task [31,32]. Therefore, we divided the experiment into two tasks, the first being a simple obstacle detection task while the second task was a complete navigation task incorporating obstacle detection and avoidance.

In the obstacle detection task, participants used the *detection mode* to locate, point at (with a laser pointer) and reach obstacles. At the same time, they were told to walk as quickly as possible while making the fewest amount of errors possible. The pointing was used to ensure that the participants had correctly detected an obstacle and not a wall. Therefore, pointing at something other than an obstacle was considered a false alarm. In the navigation task, participants were told to cross the walkway as quickly as possible, thereby avoiding any collisions with obstacles or walls while using the GSSD's *avoidance mode*.

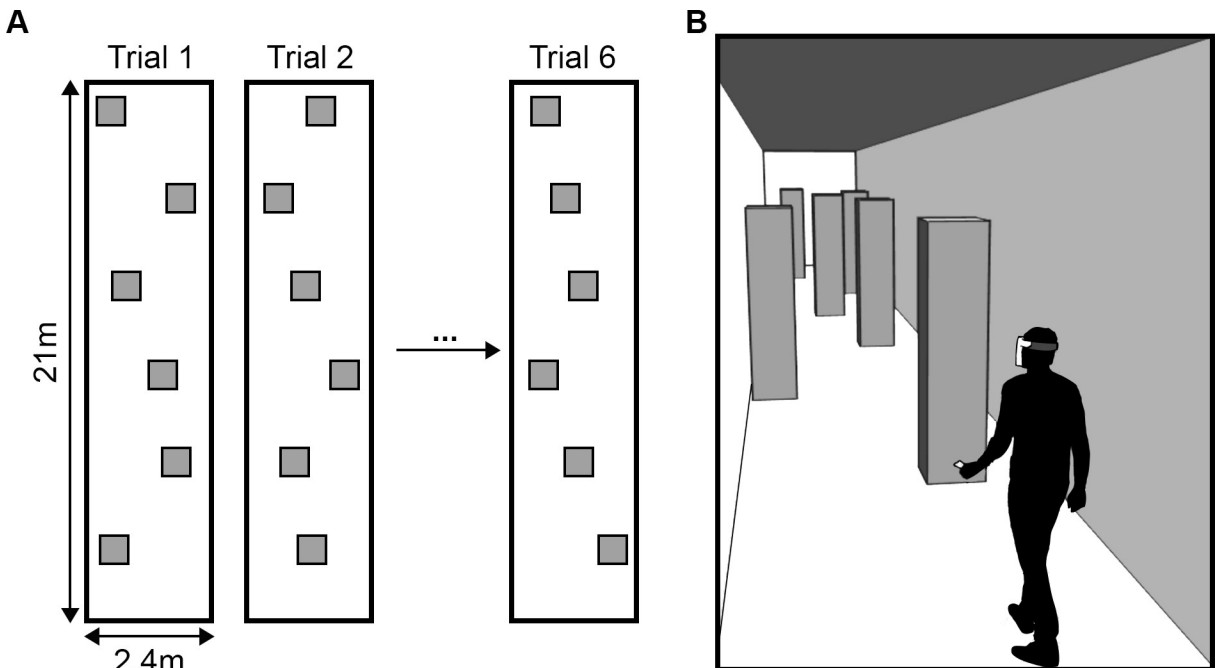

**Fig 2. Experimental design.** (A) The corridor's dimensions and examples of obstacle configurations. The obstacles are represented by grey squares. (B) Representation of the corridor with obstacles placed randomly on the horizontal axis and 3 meters apart from each other.

Both tasks consisted of six runs (12 trials in total; one trial represents a single crossing of the corridor). The participants were not aware of the number of obstacles placed along the hallway, and all obstacles changed place after each run. Every participant walked through the same six configurations for each mode (12 different configurations in total, see Fig 2A). The avoidance and detection tasks were executed on different days.

## Familiarization

Prior to the onset of the tasks, participants were asked to point to a sound source randomly located in their far space to assure they were able to localize sound sources efficiently. This was done multiple times with the sound source located in different positions and at different distances and angles from the participant, until 5 consecutively correct answers were obtained. Then, participants were explained the principles of usage for each mode, after which they were familiarized with the device. They were encouraged to interact with the device by waving their hands in front of the phone's cameras and paying attention to the auditory feedback. The participants were then placed in front of one obstacle. They were taught how to link the auditory feedback to the distance separating them from the object, how to detect an obstacle which is in the middle of the corridor, how to differentiate it from the wall and how to detect an obstacle placed against a wall. To do so, the experimenter guided the participant to the obstacle multiple times. Then, the obstacle was moved to a different location and the participant had to walk to it until it was within reach (without touching it). This forced the participants to estimate the distance using auditory cues given by the device. Then, participants were allowed to touch the obstacle to associate the auditory feedback with the tactile information. We then performed a simulation of the task with 3 obstacles placed 3 meters apart to assess the participant's understanding of the task. The familiarization process never lasted more than 30 minutes.

## Statistical analysis

Data were analyzed using JASP, an open-source graphical program for statistical analysis, developed by the University of Amsterdam. Pearson's correlation was used to evaluate if the performance was influenced by duration of blindness in the LB participants. Furthermore, two-way ANCOVAs corrected for age and sex were carried out to compare the groups' ability to detect and avoid obstacles by comparing their average performances together. Since the performances of detection and avoidance had a non-Gaussian distribution, the analyses were confirmed with the Kruskall Wallis intergroup test. Then, we compared the average crossing time between the groups using a two-way ANCOVA corrected for age and sex. This could be achieved because the average crossing time had a Gaussian distribution. The tests were done for both mode of the device. Data are expressed as mean ± SD. P values of 0.05 were considered as statistically significant.

## Results

### Obstacle detection task

All three groups detected more than 70% of the obstacles. Average detection rates for SC, LB and EB were 79.8 ± 23.5%, 73.3 ± 23.5% and 78.7 ± 23.9%, respectively (Fig 3A). A two-way ANCOVA corrected for age and sex failed to show a significant group effect for obstacle detection ($F(2,259) = 0.329$, $p = 0.72$). However, a significant effect of age was revealed by the ANCOVA ($F(1,259) = 12.557$, $p < 0.001$). There was also no significant group effect in terms of time needed to finish a run ($F(2,259) = 1.152$, $p = 0.32$) Average times to finish a run were 249 ± 97 s, 237 ± 108 s and 259 ± 114 s for SC, LB and EB, respectively (Fig 3B). We did not find, however, a significant correlation between BDI and performance of detection for LB. The analysis also failed to show an effect of sex on performance of detection and crossing time.

### Navigation task

For the navigation task, avoidance performance for EB, LB and SC were 85.1 ± 16.1%, 92.4 ± 9.9% and 85.9 ± 17.6%, respectively (Fig 4A). the two-way ANCOVA corrected for age and sex failed to show a significant group effect for obstacle detection ($F(2,193) = 2.847$, $p = 0.06$). However, there was a significant group difference for time needed to finish a run ($F$

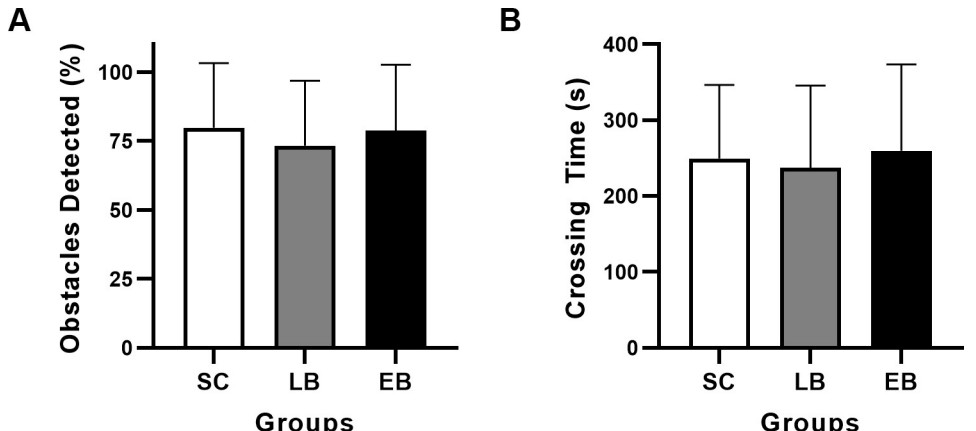

**Fig 3. Obstacle detection and average crossing time.** The average performance of detection (A) and crossing time (B) are for the six trials of the task and they are expressed in percentage of correctly detected and in seconds respectively. SC, sighted control; LB, late blind; EB, early blind.

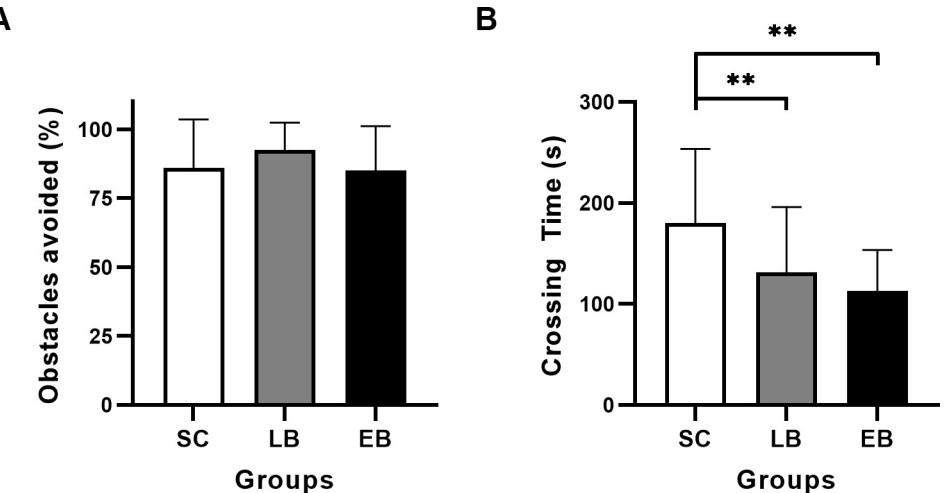

**Fig 4. Obstacle avoidance and average crossing time.** The average performance of avoidance (A) and crossing time (B) are for the six trials of the task and they are expressed in percentage of correctly detected and in seconds respectively. Significant differences are indicated by asterisks (** = p < 0.01). SC, sighted control; LB, late blind; EB, early blind.

(2,193) = 23.394, p < 0.01). Average times to finish a run were 180 ± 74 s, 132 ± 64 s and 113 ± 41 s for SC, LB and EB, respectively (Fig 4B). Post-hoc t-tests with Bonferroni correction revealed that EB (t(154) = -6.328, p < 0.01) and LB (t(142) = -4.488, p < 0.01) were significantly faster than SC to cross the corridor. We did not find, however, a significant correlation between BDI and performance of avoidance for LB. The analysis also failed to show an effect of sex and age on performance of avoidance and crossing time.

## Discussion

In this pilot study, we explored the potential and usability of the GSSD, a newly developed sensory substitution device for the blind. This application was designed to offer a simple visual-to-auditory signal that requires little training by using horizontally spatialized sounds to convey spatial data during navigation. Our data show that the GSSD allows blind and blindfolded sighted participants to detect, reach and avoid obstacles in a laboratory environment. Performance in both detection and navigation tasks exceeded 70% after a minimal amount (30 minutes) of training. These data seem to suggest that horizontally spatialized sounds that convey strict information about the location of obstacles could be sufficient to support navigation around obstacles in daily travels of those deprived of visual input. However, it is important to note that visually impaired and blind individuals constitute a heterogeneous population that will use and appreciate devices differently given the nature and time of acquisition of their condition. Furthermore, visually impaired individuals will seek tools that are easy to learn and that will complement, rather than blur or override, their remaining intact senses and abilities [22].

### Intergroup differences on device usage

**Detection and avoidance performances.** Since blind individuals constantly rely on tactile and auditory cues in daily travels, we hypothesized that they would outperform SC participants in both detection and navigation tasks. Contrary to our hypothesis, blind participants did not outperform their sighted counterparts in the detection task, nor in the navigation task. While

these results are seemingly against the known literature on blindness and SSDs [19], it is also known that the presumed advantage of EB over SC depends on the task and context [33]. Indeed, the similar performances of EB and SC may be explained by the simplicity of the obstacle course which used large and easily detectable obstacles. However, the fact that we did not observe group differences in the two tasks with different levels of complexity also suggests that the GSSD can be used efficiently by the three groups, irrespective of the task demands. The only significant difference in performance found is that younger participants, independently of blindness, were better at detecting obstacles than their older peers. This is consistent with what is known about the effect of age on sensory substitution navigation and sound localization [34,35].

**Difference in crossing time.**   Apart from performance, it is worth noting that blind participants (EB and LB) did achieve the same level of performance in the navigation task while being faster than SC. This was not found in the detection task. This can be explained by how both modes are used. Indeed, the *detection mode* necessitates scanning the environment using head rotation, as well as extension and flexion movements of the neck. Such movements are underdeveloped in EB [19,36] and are known to impair balance, straight-line travel and orientation [36,37]. As for the *avoidance mode*, participants received auditory inputs from multiple sound sources simultaneously, allowing them to plan their route without the need for head movements. Furthermore, it is known that multiple sound sources can improve postural control in LB and EB [38] and that they are better at localizing sounds in the periphery [39], a necessary skill to use the device in *avoidance mode*. Indeed, to advance in the obstacle course, participants had to orient themselves in order to place all sounds far enough in the periphery, free the space upwards and avoid collisions with the obstacles. Furthermore, this finding is consistent with previous behavioral and neurophysiological studies on the use of SSDs. While it has not been investigated in LB yet, it has been demonstrated that EB are more efficient in the use of SSDs [15,21,23] and that the EB brain recruits functional networks used by sighted individuals for navigation such as the visual dorsal stream and the parahippocampal gyrus while blindfolded SC do not [40].

Alternatively, our results could also be explained by the fact that it is unusual for SC to navigate blindfolded, while it is the daily experience of the other two groups [41]. Indeed, when compared with blind individuals who only benefit from the additional information given by the SSD, being blindfolded (even with the SSD) is an incapacitating and disorienting event for SC. It eliminates visual inputs that they rely upon for navigation and postural control [37,42]. This could have transcribed into increased transit times and moderation of pace for SC when deprived of vision and guided only with new auditory cues. Moreover, since bone-conducting headphones were used, participants had preserved access to regular auditory cues. Therefore, it cannot be ruled out that environmental sounds could have influenced the performance of the blind. Indeed, it is known that blind individuals are trained to use environmental sounds to integrate spatial information and to detect obstacles with passive or active echolocation [43–45]. However, our experiment was conducted in a quiet environment and we verified that participants did not use active echolocation. Furthermore, passive echolocation relies on environmental sounds [46]. Considering the quiet environment and that no significant difference was found in detection performances per se, it seems unlikely that passive echolocation is the sole reason for the faster pace of the blind. In contrast, using environmental sounds simultaneously with the GSSD could benefit the user. According to principles of ergonomic design for devices used by the blind, it is necessary that the feedback given by the device does not interfere nor undermine the use of other senses and abilities already developed and efficient for safe navigation. In other words, devices should complement the blinds' abilities rather than keep them from using these [22]. Therefore, the purpose of our GSSD is to take advantage of the

information the user has access to and not restrict it to the input of the device. For these reasons, bone-conducting headphones that keep the ears free were chosen for the experiment and for the use of our GSSD.

## Sensory substitution devices

The potential and usability of sensory substitution devices for spatial navigation have been amply discussed in recent years. There is a consensus that SSDs can expand the blind's perception of the environment and give them a "visual-like" experience of space. However, it is difficult to properly judge what design has the most potential due to the numerous different experimental designs and the general lack of testing in ecological (or "real-world") settings [19].

A study on the use of the TDU in a navigation task showed that both congenital blind individuals (CB) and sighted controls were able to detect and avoid obstacles in a life-size obstacle course [15]. The authors demonstrated that CB were better than their sighted counterparts at using the TDU to navigate with performances very similar to ours for obstacle detection, but inferior than ours in obstacle avoidance. However, this could be explained by the complexity of their testing set-up which included widely different types of obstacles, whereas we only presented identically sized obstacles. Nevertheless, their participants trained for several hours before doing the experiment [15] while ours trained only for 30 minutes.

A recent study showed that using the TDU, and even the vOICe, could help individuals transfer information from a virtual map to a real environment [47]. It has been argued that the vOICe can achieve superior spatial resolution than the TDU [8]. Nevertheless, since the vOICe scans the field of view only every 2 seconds, it lacks the temporal resolution of others, such as our GSSD. Consequently, no real navigation and obstacle avoidance testing has been done with the vOICe since the time delay precludes a real time perception of the surroundings. However, several studies demonstrated that with sufficient training, blind and blindfolded sighted individuals can use the vOICe to locate and identify immobile objects in space [20,48,49]. Nonetheless, participants of these studies trained intensively for 15 hours [20], 40 hours [49] and even for several months [50].

Moreover, the vOICe and the TDU do not give direct information about depth which is crucial for navigation [11,51]. Indeed, these devices require the user to move the head to grasp the environment within its visual-field and then to analyze the area of the visual-field occupied by the object to extract distance information [52]. This makes the understanding of an already complex signal even more difficult [22]. This is a quasi-automatic process under normal vision, consequently unknown and laborious to understand for EB individuals. This has been demonstrated with EB individuals unable to perceive the Ponzo illusion with a visual-to-auditory SSD, while it was perceived by SC and LB [39].

As for the minimalist SSD based upon the "point-to-distance" concept, the Eyecane has been studied in many different obstacle avoidance and navigation studies. These studies all demonstrated that visually impaired individuals can detect and avoid obstacles [24], be efficient in a pathfinding task [23] as well as in translating information from a virtual maze to a real environment. [21] after a minimum amount of training. This suggests that simplifying the feedback can help the blind to navigate by focusing on the navigation task, rather than spending resources on the interpretation of a complex signal.

Perhaps the most alike SSD to the GSSD in terms of sonification, is the SoV [53]. Indeed, their sonification strategy incorporated similar water droplets or "bubbles" sound effects and binaural sound differences as part of their feedback. However, their sonification strategy is used to convey spatial information relevant for both navigation and object identification. The

SoV conveys all points and edges of obstacles so participants can analyze the object shape and volume in space, requiring several hours of training.

Thus, our results suggest that the GSSD helps visually impaired individuals to navigate efficiently between obstacles with less training than other previously and currently tested SSDs. Such efficiency is achieved by restricting the transmitted information to spatial cues relevant to avoiding collisions (distance and direction). Visually impairments individuals give more importance to their safety and expect navigational aids to detect obstacles and give warning so they can be safely guided towards their destination [54], a goal that is accomplished with the GSSD. Moreover, an aspect of the GSSD that we have not explored in the experiment is the possibility to modify important settings of the GSSD to better suit individual needs and tasks. For instance, the user can personalize his/her experience by choosing between a variety of sounds. The user can also adjust the horizontal and vertical field of view of the system, as well as the detection range of the system, to adequately deal with closed narrow and open environments. We therefore believe that our device can bring better subjective appreciation than other SSDs.

## Conclusion

The present study shows the GSSD's potential towards easier, safer and more autonomous navigation for the visually impaired by using horizontally spatialized sounds as sole feedback. A simple and strictly relevant spatial feedback might prove to be easy to use and to allow for faster spatial learning. Indeed, we suggest that the GSSD can minimize cognitive load for the user in a similar fashion than the EyeCane [23], while giving more numerous and simultaneous cues about the environment. It is also important to note that the GSSD is designed not to replace, but rather to complement the blind's tools and abilities such as the white cane and echolocation. Future work should focus on the use of the GSSD in more complex and ecological environments.

## Supporting information

**S1 Dataset.**
(XLSX)

## Acknowledgments

The authors wish to thank Chuck Knowledge and Danny Bernal from SignalGarden for developing the GSSD software.

## Author Contributions

**Conceptualization:** Samuel Paré, Maxime Bleau, Ismaël Djerourou, Vincent Malotaux, Ron Kupers, Maurice Ptito.

**Data curation:** Samuel Paré, Maxime Bleau, Ismaël Djerourou, Ron Kupers, Maurice Ptito.

**Formal analysis:** Samuel Paré, Maxime Bleau, Ismaël Djerourou.

**Funding acquisition:** Ron Kupers, Maurice Ptito.

**Investigation:** Samuel Paré, Maxime Bleau, Ismaël Djerourou, Vincent Malotaux, Ron Kupers, Maurice Ptito.

**Methodology:** Samuel Paré, Maxime Bleau, Ismaël Djerourou, Vincent Malotaux, Ron Kupers, Maurice Ptito.

**Project administration:** Ron Kupers, Maurice Ptito.

**Resources:** Vincent Malotaux, Ron Kupers, Maurice Ptito.

**Software:** Vincent Malotaux, Ron Kupers, Maurice Ptito.

**Supervision:** Ron Kupers, Maurice Ptito.

**Validation:** Samuel Paré, Maxime Bleau, Ron Kupers, Maurice Ptito.

**Visualization:** Samuel Paré, Maxime Bleau, Ismaël Djerourou, Maurice Ptito.

**Writing – original draft:** Samuel Paré, Maxime Bleau, Ismaël Djerourou.

**Writing – review & editing:** Samuel Paré, Maxime Bleau, Ron Kupers, Maurice Ptito.

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
