## [Decision Letter · Decision Letter 0]

5 Oct 2020

PONE-D-20-25379

Spatial navigation with horizontally spatialized sounds in early and late blind individuals

PLOS ONE

Dear Dr. Ptito,

Thank you for submitting your manuscript to PLOS ONE. After careful consideration, we feel that it has merit but does not fully meet PLOS ONE’s publication criteria as it currently stands. Therefore, we invite you to submit a revised version of the manuscript that addresses the points raised during the review process.

Please carefully consider the detailed reviews and use them in revising your manuscript.

We look forward to receiving your revised manuscript.

Kind regards,

Thomas A Stoffregen, PhD

Academic Editor

PLOS ONE

Journal Requirements:

2.Thank you for stating the following financial disclosure:

 [The funders had no role in study design, data collection and analysis, decision to publish, or preparation of the manuscript.].

3.We note that you have indicated that data from this study are available upon request. PLOS only allows data to be available upon request if there are legal or ethical restrictions on sharing data publicly. For more information on unacceptable data access restrictions, please see http://journals.plos.org/plosone/s/data-availability#loc-unacceptable-data-access-restrictions.

4. Please change "female” or "male" to "woman” or "man" as appropriate, when used as a noun."

Additional Editor Comments (if provided):

Two highly qualified Reviewers have offered detail comments. Based on their input, I am requesting a Major Revision. Both Reviewers point out the need better to review relevant prior work. Each Reviewer also offers specific suggestions about details of the manuscript.

Reviewers' comments:

Reviewer's Responses to Questions

**Comments to the Author**

1. Is the manuscript technically sound, and do the data support the conclusions?

Reviewer #1: Yes

Reviewer #2: Partly

2. Has the statistical analysis been performed appropriately and rigorously? 

Reviewer #1: Yes

Reviewer #2: Yes

3. Have the authors made all data underlying the findings in their manuscript fully available?

Reviewer #1: Yes

Reviewer #2: Yes

4. Is the manuscript presented in an intelligible fashion and written in standard English?

Reviewer #1: Yes

Reviewer #2: Yes

5. Review Comments to the Author

Reviewer #1: Spatial navigation with horizontally spatialized sounds in early and late blind individuals

Pare, et al.

An experiment tests a novel sensory substitution device (SSD) for the blind which provides horizontal and depth location information for objects via sound. The authors claim that the device is novel by virtue of providing horizontal location using interaural sound cues. Three subject groups were tested: early blind, late blind, and blindfolded sighted subjects on their ability to determine the location of obstructions and to navigate through a set of obstructions. All three groups show some success at both tasks with some differences observed between the groups.

The authors may have designed a very unique and promising SSD which may provide more usable information for guidance. Unfortunately, the write-up does not provide enough information about the full range of existing SSDs, so it is impossible for me to evaluate precisely how novel their device is, and whether it truly provides more usable information. Have no other SSD devices provided horizontal location information of obstacles using interaural cues? It is impossible to tell the answer to this question based on the cursory description of existing devices. Furthermore, without a more close comparison between performance with this device and that with others, it is impossible to assess the GSSD’s success. It actually may be that the GSSD is easier to use (and learn) and is more effective than what is currently available. However, there is not enough description of the earlier findings (e.g., testing methods and tasks; overall performance; learning curves) for a critical comparison to be made. For these reasons, I must recommend rejection of the current manuscript. However, I hope the authors can rework their paper to address these issues.

More minor comments:

p. 4, line 66 – “guidance-SSD” has not been defined yet, the claim that it can “minimize strain” is meaningless at this point in the paper.

p. 4 – much more discussion is needed of exactly how other sound-based SSDs (e.g., vOICe) code for horizontal position. Are there any existing SSDs that use interaural cues?

p. 4, line 80 – If the main purpose of the project is to determine whether the GSSD might be more effective, then it is unclear exactly why the three subject groups are being tested. Also, given what seems to be this main purpose, the hypotheses listed here aren’t well motivated.

p. 7, line 127– It is unclear what the basic signal being used is like. Is it pure tone? And what are the average frequency and intensity values like? How have these decisions been made? Making an audio example available somewhere online would help readers immensely.

p. 7, line 138— How are the widths of objects conveyed? Is there some type of interaural information available for this?

Results – It should be made clear what performance values should be expected if the device is successful. These values should allow comparison to those of results from other SSD research (e.g., from vOICe experiments).

p. 11, line 231 – It should be made clear here that these percentage values are calculated based on the percentage of obstacles not touched by the subjects during the avoidance task.

p. 12, line 252 – This conclusion statement needs some type of context – preferably relative to prior SSD research.

p. 12-13 – The differences between subject groups is really a secondary concern and doesn’t really warrant this much discussion in a short report. The reader is most interested in how this new device fairs relative to other SSDs.

Reviewer #2: Overview

The authors have developed a smartphone-based sensory substitution device (“SSD”), which enables visually impaired persons to navigate their routes by auditory feedback. In this paper, the authors work on demonstrating the feasibility of the SSD of three groups: early blind, late blind, and sighted participants.

The paper presents a valuable road map for further discussion and device development with two points. First, the SSD is designed that bone-conducting headphones allow blind persons to use both their own auditory sense and feedback from the SSD, so that they can detect and avoid non-sounding obstacles placed three meters apart from each other within an experimental corridor. Second, the SSD does not require long-term training to detect obstacles and relate the obstacles to the auditory feedback signals.

However, the paper does not provide the readers with sufficient information on the SSD specifications and the results of navigation experiments. I recommend that the authors should add details that logically support their arguments and make it more convincing to the readers.

Therefore, I would suggest a major revision.

My specific comments on the paper are as follows.

L44-45: Previous studies must be referred to support this statement.

L49-50: This is about detecting obstacles placed on the vertical direction, though the paper mainly focuses on using horizontal spatial cues. This sentence is not related to the context of the paper.

L67-69: It does not seem that the experimental design, results, and discussion are consistent with the goal of the study. From the experiment results and discussion alone, it is difficult to figure out how effectively the SSD (i.e., horizontally spatialized sounds feedback) provides the users with the spatial configuration environment.

L75-77: No reason is given to make a manipulative concept of “cognitive map” the basis to meet the goal mentioned on L78-79. There is no need for the reference of a cognitive map, the role of which is not mentioned in the result and discussion of the experiment section.

L81: It does not seem that the experiment is designed to prove that hypothesis. Especially, the authors must give enough data to convince the readers that the developed SSD needs shorter training than conventional devices to perform obstacle detection and avoidance tasks.

L88 (Participants and Ethics): Hearing abilities of the participants should be described.

L115 (Apparatus): For the readers’ better understanding of new SSD, it is necessary to provide more detailed specifications of the auditory feedback system developed exclusively for this study, in particular:

- Latency time that the SSD holds, from detecting obstacles to outputting horizontal sound feedback,

- Maximum number of obstacles the SSD can respond and create auditory feedback, and

- Quantitative evidence that describes how the users can estimate the distance to an obstacle accurately by using the combination of three acoustic components.

L141: For the readers’ better understanding, this should be supported with quantitative data, such as how many objects people can hear and distinguish.

L164 (Tasks): For experimenting the effect of auditory feedback from the SSD on the obstacle detection/avoidance tasks, it is critical to make sure that the participants cannot use other auditory cues available in the experiment corridor, possibly by adding earplugs and/or earmuffs to shut off background noise.

L187 (Familiarization): To convince that new SSD gives shorter training than conventional SSDs to learn how to use, it is necessary to compare the required time for training between them.

L218-219: More thorough, detailed discussion is necessary.

L234-235: It is necessary to discuss the possibilities that blind participants can use ambient sound available in the experiment location. This will effectively support the result that EB and LB finished a run faster than SC.

L254: Experiment results do not give sufficient data-based evidence to support the point. Quantitative data, such as the comparison with other SSDs in terms of time required for training, is necessary to convince the readers that the new SSD requires little training.

L273-276: It seems that authors overvalue the data they have obtained from the experiment.

L279-280: This is a fascinating statement. With prior studies on the plasticity of brain function of EB, the authors argue that the different duration to complete tasks among three groups is based on the nerve system. However, this argument does not well-support the quick task completion of LB. Also, the duration for task accomplishment is not directly related to the different nerve system functions.

L290-293: Here the authors emphasize that SC, who relies typically on the vision for higher cognitive spatial processing, may restrict the capability when they are “deprived of vision” during runs. However, no quantitative data are presented from the experiment to support the direct relationship between short duration of completing navigation tasks and higher cognitive spatial processing. Discussion about this may help the readers better understand why the blind groups and the sighted group were different in the time duration.

L295-296: Quantitative data is necessary to present how many sound sources the participants perceived during each run.

L328: Experiment results of this study may be too weak to support this conclusion because there are no control conditions that limit the participants to hearing the environmental sound.

L329: To help the readers understand the influence of visual experience on blind persons, this sentence needs clarity. It is unclear for the readers to find whether or not there is an influence of visual experience.

6. PLOS authors have the option to publish the peer review history of their article (what does this mean?). If published, this will include your full peer review and any attached files.

Reviewer #1: No

Reviewer #2: No

---

## [Author Response · Author response to Decision Letter 0]

16 Nov 2020

Response to Reviewers’ comments

Reviewer #1 

An experiment tests a novel sensory substitution device (SSD) for the blind which provides horizontal and depth location information for objects via sound. The authors claim that the device is novel by virtue of providing horizontal location using interaural sound cues. Three subject groups were tested: early blind, late blind, and blindfolded sighted subjects on their ability to determine the location of obstructions and to navigate through a set of obstructions. All three groups show some success at both tasks with some differences observed between the groups.

The authors may have designed a very unique and promising SSD which may provide more usable information for guidance. Unfortunately, the write-up does not provide enough information about the full range of existing SSDs, so it is impossible for me to evaluate precisely how novel their device is, and whether it truly provides more usable information. Have no other SSD devices provided horizontal location information of obstacles using interaural cues? It is impossible to tell the answer to this question based on the cursory description of existing devices. Furthermore, without a more close comparison between performance with this device and that with others, it is impossible to assess the GSSD’s success. It actually may be that the GSSD is easier to use (and learn) and is more effective than what is currently available. However, there is not enough description of the earlier findings (e.g., testing methods and tasks; overall performance; learning curves) for a critical comparison to be made. For these reasons, I must recommend rejection of the current manuscript. However, I hope the authors can rework their paper to address these issues.

We thank the reviewer for this pertinent comment. Our GSSD was designed with the purpose to have a system that requires less effort and cognitive load for the user. The GSSD gives direct information about positioning and depth, and requires minimal processing from the user. Therefore, we added in the introduction (see lines 54 to 77) and Discussion (pages 18 - 21) sections a more detailed description of the other existing SSDs.

More minor comments:

p. 4, line 66 – “guidance-SSD” has not been defined yet, the claim that it can “minimize strain” is meaningless at this point in the paper.

We changed the paragraph (lines 91-102) to define the need for simpler SSDs and added the example of the EyeCane. We formulated our hypothesis that by giving less but exclusively relevant information, the device provides simple feedback which enhances navigation abilities. Next, we introduced our new application that uses horizontally spatialized sounds to “guide” the user through obstacles, hence the name “Guidance-SSD” (GSSD).

p. 4 – much more discussion is needed of exactly how other sound-based SSDs (e.g., vOICe) code for horizontal position. Are there any existing SSDs that use interaural cues?

As we wrote above, we added rather lengthy descriptions of other SSDs and how they work, thereby focusing on the best studied SSDs for navigation (for a review on the subject see Chebat et al., 2018). We added the following text to the Introduction:.

“The vOICe is one of the best studied visual-to-auditory (VTA) SSD. The camera of the device scans its field of view from left to right, thereby offering momentary “snapshots” of the environment in the form of sound cues. The vOICe informs the user about the vertical and horizontal positioning of objects, as well as brightness of the environment. Vertical and horizontal positioning are indicated by the frequency and the length of the sound, respectively, whereas brightness is indicated by differences in amplitudes of the sound oscillations. The vOICe demands the user to analyze multiple spectral cues, with a two second delay between each scan, to extract important information, detect and identify objects [11].

The Tongue Display Unit (or TDU) is a tactile-to-vision (TTV) SSD capable of transmitting images to the tongue in the form of electrotactile pulses. The TDU is composed of a tongue array consisting of 400 small circular electrodes arranged in a 20x20 matrix, a computer and a webcam. Every time an object enters within the visual field of the camera, the visual image is translated into electrotactile pulses that are transmitted to the tongue through the electrode array. The obstacles are thus ‘drawn’ with electrical current on the tongue in real time from the images provided by the camera [15-17].

The Sound of Vision (SoV) is a more recently developed VTA SSD. The SoV provides combined audio and tactile feedback by using multiple cameras and depth sensors that are worn on the forehead and which are connected to a laptop stowed in a backpack and worn by the user. The system informs the user of obstacles positioning with vibrations to the abdomen through a haptic belt. Then, the SoV conveys depth information (or overall 3D objects’ shape) to the user by translating all 3D points into binaural sound effects of “popping bubbles” that will be modulated in loudness and pitch for proximity and elevation respectively [14].”

p. 4, line 80 – If the main purpose of the project is to determine whether the GSSD might be more effective, then it is unclear exactly why the three subject groups are being tested. Also, given what seems to be this main purpose, the hypotheses listed here aren’t well motivated.

Age of onset of blindness has an important effect on compensatory neuroplastic changes (for a review, see Kupers and Ptito, Neurosci Biobehav. Rev, 2014). Congenitally blind and late-onset blind individuals use different navigational strategies, and differ in auditory capacities, and in cognitive processes. Late and early-onset blindness pose completely different challenges to the individual which can greatly affect the way they use a SSD. In order to test finetune a novel SSD, it is crucial to evaluate its potential in both early and late-onset blindness. 

p. 7, line 127– It is unclear what the basic signal being used is like. Is it pure tone? And what are the average frequency and intensity values like? How have these decisions been made? Making an audio example available somewhere online would help readers immensely.

The audio signal is customizable to the user. We used a raindrop sound since fluid sounds such as water droplets and bubbles are easily identifiable and localizable due to their natural and regular occurrence (Doel, ACM Transactions on Applied Perception (TAP), 2005). We added a more precise description of the auditory signal in the revised manuscript (see lines 169 to 174). 

p. 7, line 138— How are the widths of objects conveyed? Is there some type of interaural information available for this?

The device does not provide direct width information, because it would constitute additional information that could complexify the signal. Thus, the sound source always conveys information about a single point in space. However, this information can be used, depending on the mode, to extract such information if needed. In detection mode, the user can estimate the width of objects by scanning the object and comparing the point in space where it is first and last detected. They thus have to estimate the width using the angle information of their scan with the distance heard by the object. In the avoidance mode, no information can be deduced on the width of the object. 

Results – It should be made clear what performance values should be expected if the device is successful. These values should allow comparison to those of results from other SSD research (e.g., from vOICe experiments).

We have added this now to the manuscript (see lines 110 to 116):

“To test this hypothesis, we investigated if participants could detect and avoid obstacles in a life-size obstacle course using this new application to guide their movements. We included early blind (EB), late blind (LB) and blindfolded sighted control subjects (SC) in a detection and avoidance task. We hypothesized that participants would be able to perform both tasks above chance level with a minimum amount of training. Moreover, given the established spatial auditory capacities of the blind [10, 20], we supposed that both blind groups would outperform SC and that EB would be better than LB.”

p. 11, line 231 – It should be made clear here that these percentage values are calculated based on the percentage of obstacles not touched by the subjects during the avoidance task.

Performance in the avoidance task is calculated as the percentage of obstacles avoided (i.e. not touched by the participant). A performance of 50% would mean that a participant touched or collided with three obstacles out of six in a trial. We also reformulated “performance” to “avoidance performance” in the results section of the navigation task.

p. 12, line 252 – This conclusion statement needs some type of context – preferably relative to prior SSD research.p. 12, line 252 – This conclusion statement needs some type of context – preferably relative to prior SSD research.

Indeed, we added a context to introduce this statement. Furthermore, comparisons with other SSDs were added in the discussion to relate to this statement (see pages 18 to 21).

p. 12-13 – The differences between subject groups is really a secondary concern and doesn’t really warrant this much discussion in a short report. The reader is most interested in how this new device fairs relative to other SSDs.

We agree with the reviewer and we adapted the Discussion section accordingly. More specifically, we added more details on how the new GSSD compares to existing SSDs in terms of ease of use, training time and navigational performance (see pages 18 to 21). However, we kept a shortened discussion on group differences in performance since many behavioral differences between EB and LB exist in the literature, especially with respect to auditory capacities and navigational strategies. For instance, EB are better at analyzing spectral cues and binaural cues than LB. EB tend to use a more egocentric strategy while LB rather use a mixture of egocentric and allocentric strategies. 

Reviewer #2: Overview

The authors have developed a smartphone-based sensory substitution device (“SSD”), which enables visually impaired persons to navigate their routes by auditory feedback. In this paper, the authors work on demonstrating the feasibility of the SSD of three groups: early blind, late blind, and sighted participants. The paper presents a valuable road map for further discussion and device development with two points. First, the SSD is designed that bone-conducting headphones allow blind persons to use both their own auditory sense and feedback from the SSD, so that they can detect and avoid non-sounding obstacles placed three meters apart from each other within an experimental corridor. Second, the SSD does not require long-term training to detect obstacles and relate the obstacles to the auditory feedback signals. However, the paper does not provide the readers with sufficient information on the SSD specifications and the results of navigation experiments. I recommend that the authors should add details that logically support their arguments and make it more convincing to the readers. Therefore, I would suggest a major revision. My specific comments on the paper are as follows.

L44-45: Previous studies must be referred to support this statement.

We added three references that support this statement: Patla AE., Advances in psychology, 1991; Patla & al., Journal of Experimental Psychology: Human Perception and Performance, 1991; McFadyen & al., Experimental brain research, 2007

L49-50: This is about detecting obstacles placed on the vertical direction, though the paper mainly focuses on using horizontal spatial cues. This sentence is not related to the context of the paper.

We thank the reviewer for this comment. We have clarified that this statement is about identifying the present limitations that face blind people during navigation. We then specified that the GSSD is used to detect all tangible obstacles, regardless of their height (i.e. knee-high obstacles, hanging obstacles, or tall obstacles), and that the auditory feedback will allow the user to “localize” the objects on the horizontal plane to adjust his/her path of travel around the obstacle (regardless of its height). By doing so, the GSSD reduces the limitations faced by the user when only using the long cane (line 105).

L67-69: It does not seem that the experimental design, results, and discussion are consistent with the goal of the study. From the experiment results and discussion alone, it is difficult to figure out how effectively the SSD (i.e., horizontally spatialized sounds feedback) provides the users with the spatial configuration environment.

This is a pilot study to determine if the new GSSD allows obstacle circumvention. We provided more information on the device to ease the understanding of its functionalities (see page 9). The results show that participants were able to avoid obstacles and navigate between them efficiently with the GSSD. We agree that the Results and Discussion focused too heavily on intergroup differences rather than on testing this principle. Therefore, we changed the discussion to clarify how the device is used in the different tasks and how it compares to other SSDs (see pages 18 to 21).

The fact that all 3 groups were able to use the device in a proficient manner suggests that it is simple to use in the context of our tasks.

L75-77: No reason is given to make a manipulative concept of “cognitive map” the basis to meet the goal mentioned on L78-79. There is no need for the reference of a cognitive map, the role of which is not mentioned in the result and discussion of the experiment section.

We agree that the concept of “cognitive map” may be irrelevant in this section. Therefore, we reformulated the sentence (lines 107 to 109) : 

“Therefore, we hypothesize that when confronted with obstacles in their path of travel, participants using the application should have enough information to be “guided” through obstacles, hence the GSSD name.”

L81: It does not seem that the experiment is designed to prove that hypothesis. Especially, the authors must give enough data to convince the readers that the developed SSD needs shorter training than conventional devices to perform obstacle detection and avoidance tasks.

We thank the reviewer for this comment. Therefore, the discussion part of the manuscript was adapted by adding more details about how our new GSSD compares to other SSDs in ease of use, training time and in navigation task performances (see pages 18 to 21).

L88 (Participants and Ethics): Hearing abilities of the participants should be described.

We did not evaluate the hearing abilities of the participants directly. However, when recruiting the participants we controlled for any associated neuropathy (including hearing disorders) that could influence the participants behavior and results. Furthermore, during familiarization, participants were asked to point to a far located sound source to assure they were able to localize sound in far space. All participants were successful and precise at pointing the sound sources. We added this part of the training in the revised manuscript (lines 245 to 249). 

L115 (Apparatus): For the readers’ better understanding of new SSD, it is necessary to provide more detailed specifications of the auditory feedback system developed exclusively for this study, in particular: latency time that the SSD holds, from detecting obstacles to outputting horizontal sound feedback, maximum number of obstacles the SSD can respond and create auditory feedback, and quantitative evidence that describes how the users can estimate the distance to an obstacle accurately by using the combination of three acoustic components.

The SSD offers a real-time three-dimensional mapping of the environment with its spatial auditory feedback. The avoidance mode is designed to detect simultaneously every distinguishable object in the camera’s field of view. However, since the obstacle course had an obstacle every 3 meters (which is the maximum range the device), participants could only detect both walls and one obstacle at the same time. Therefore, only three tangible objects could be detected at once (see lines 186-187). No direct results on distance estimation were taken. However, the process of obstacle circumvention requires an individual to estimate distance correctly. Indeed, it requires the individual to distance his/her body far enough from the object to avoid it safely (see Kolarik et al., Experimental brain research, 2016). Therefore, in the context of navigation, correct obstacle avoidance is based upon correct distance estimation. We discussed this matter in the revised manuscript (see lines 212-215). 

L141: For the readers’ better understanding, this should be supported with quantitative data, such as how many objects people can hear and distinguish.

In the case of this study, participants could detect both walls and an obstacle at once for a maximum of three objects detected simultaneously in this specific experimental set-up. (should be interesting to test how many sound sources participants can distinguish). This has been added in the text (see lines 186-187).

L164 (Tasks): For experimenting the effect of auditory feedback from the SSD on the obstacle detection/avoidance tasks, it is critical to make sure that the participants cannot use other auditory cues available in the experiment corridor, possibly by adding earplugs and/or earmuffs to shut off background noise.

We discussed the effect of bone-conducting headphones and environmental sounds in the manuscript (see lines 375 to 392). The reason we choose to keep the participants’ ears free is because environmental sounds are an important source of navigational information for the blind. According to principles of ergonomic design for devices used by the blind, it is necessary that the feedback given by the device does not interfere with the use of other senses and abilities already developed and efficient for safe navigation. In other words, devices should complement the blinds’ abilities rather than keep them from using these said abilities. Therefore, we opted for bone conduction of the sound information. Moreover, our goal was to assess the device in natural condition under which participants are free to use other auditory feedback. Since our testing environment was quiet, it is unlikely that participants have used environmental sound cues.

L187 (Familiarization): To convince that new SSD gives shorter training than conventional SSDs to learn how to use, it is necessary to compare the required time for training between them.

See also our response to reviewer 1 for a similar comment. We have added more details about how our new GSSD compares to other SSDs in terms of ease of use, training time and in navigational performance (see pages 18 to 21).

L218-219: More thorough, detailed discussion is necessary.

We have thoroughly restructured the Discussion section and mentioned two studies (Levy-Tzedek S et al., PloS one, 2016; Dobreva et al., Journal of neurophysiology, 2011) that demonstrated the influence of age on the use of SSD for navigation and for sound localization (see lines 346 to 348).

L234-235: It is necessary to discuss the possibilities that blind participants can use ambient sound available in the experiment location. This will effectively support the result that EB and LB finished a run faster than SC.

Thank you for pointing this out. We have now added this to the Discussion (see lines 375 to 392).

L254: Experiment results do not give sufficient data-based evidence to support the point. Quantitative data, such as the comparison with other SSDs in terms of time required for training, is necessary to convince the readers that the new SSD requires little training.

We added to the Discussion more details about how our new GSSD compares to other SSDs in terms of ease of use, training time and in navigation task performance (see pages 18 to 21). 

L273-276: It seems that authors overvalue the data they have obtained from the experiment.

As mentioned above, we have thoroughly restructured the Discussion and reduced the emphasis on group differences (see lines 346 to 348).

L279-280: This is a fascinating statement. With prior studies on the plasticity of brain function of EB, the authors argue that the different duration to complete tasks among three groups is based on the nerve system. However, this argument does not well-support the quick task completion of LB. Also, the duration for task accomplishment is not directly related to the different nerve system functions.

We have modified the text in accordance with the reviewer’s comment concerning LB and EB (see lines 364-367)

L290-293: Here the authors emphasize that SC, who relies typically on the vision for higher cognitive spatial processing, may restrict the capability when they are “deprived of vision” during runs. However, no quantitative data are presented from the experiment to support the direct relationship between short duration of completing navigation tasks and higher cognitive spatial processing. Discussion about this may help the readers better understand why the blind groups and the sighted group were different in the time duration.

We agree with the reviewer and we have adapted this part of the discussion by focusing on the purported negative effects blindfolding may have in sighted individuals, especially for navigation and for postural control. We also added 3 references (Schwesig et al., European journal of ophthalmology, 2011; Ribadi et al., Adapted Physical Activity Quarterly, 1987; Heller et al., Psychology press, 2013) that support this statement (see lines 369 to 375).

L295-296: Quantitative data is necessary to present how many sound sources the participants perceived during each run.

Since an obstacle was placed every 3 meters (which is the maximum range the device), participants could only detect both walls and one obstacle at the same time. Therefore, only three tangible objects could be detected at once. Furthermore, since the SSD covers the entire width of the corridor, every participant detected the same number of obstacles and therefore was presented with as many sound sources as the other participants (lines 186-187). 

L328: Experiment results of this study may be too weak to support this conclusion because there are no control conditions that limit the participants to hearing the environmental sound.

Indeed, since participants had their ears free, we cannot totally exclude the possibility that they could have used other sound sources. We discussed this issue in lines 375 - 392 and we also reformulated the Conclusion (see lines 457 to 465). 

L329: To help the readers understand the influence of visual experience on blind persons, this sentence needs clarity. It is unclear for the readers to find whether or not there is an influence of visual experience.

Thank you for this suggestion. The sentence has been removed.

---

## [Editor Report · Decision Letter 1]

8 Feb 2021

Spatial navigation with horizontally spatialized sounds in early and late blind individuals

PONE-D-20-25379R1

Dear Dr. Ptito,

We’re pleased to inform you that your manuscript has been judged scientifically suitable for publication and will be formally accepted for publication once it meets all outstanding technical requirements.

Kind regards,

Thomas A Stoffregen, PhD

Academic Editor

PLOS ONE
---

## [Editor Report · Acceptance letter]

9 Feb 2021

PONE-D-20-25379R1 

Spatial navigation with horizontally spatialized sounds in early and late blind individuals 

Dear Dr. Ptito:

I'm pleased to inform you that your manuscript has been deemed suitable for publication in PLOS ONE. Congratulations! Your manuscript is now with our production department. 

Kind regards, 

on behalf of

Dr. Thomas A Stoffregen 

Academic Editor

PLOS ONE